# Beyond CTLA-4 and PD-1 Inhibition: Novel Immune Checkpoint Molecules for Melanoma Treatment

**DOI:** 10.3390/cancers15102718

**Published:** 2023-05-11

**Authors:** Dimitrios C. Ziogas, Charalampos Theocharopoulos, Panagiotis-Petros Lialios, Dimitra Foteinou, Ioannis-Alexios Koumprentziotis, Georgios Xynos, Helen Gogas

**Affiliations:** First Department of Medicine, Laiko General Hospital, School of Medicine, National and Kapodistrian University of Athens, 11527 Athens, Greece; hartheoch@gmail.com (C.T.); lialiospeter@gmail.com (P.-P.L.); foteinoudim@gmail.com (D.F.); giannhskmpr@gmail.com (I.-A.K.); george.xynos@gmail.com (G.X.)

**Keywords:** immune checkpoints, immune checkpoint inhibitors, melanoma, immunotherapy resistance, relatlimab, LAG-3, TIGIT, TIM-3, VISTA

## Abstract

**Simple Summary:**

Although the incorporation of immune checkpoint inhibitors (ICIs) in melanoma treatment has significantly improved prognosis for patients with advanced disease, a substantial proportion of patients have limited clinical benefit from ICIs’ administration. Except for the well-studied PD-1 and CTLA-4 immune checkpoints, the expression of several other molecules has been associated with immune resistance and T-cell exhaustion. In this overview, we present the functional characteristics of presently described immune checkpoints, including LAG-3, TIGIT, TIM-3, VISTA, IDO1/IDO2/TDO, CD27/CD70, CD39/73, HVEM/BTLA/CD160 and B7-H3, with an emphasis on early clinical and preclinical data over the novel therapeutic agents that target these molecules. The aim of our review is to enrich the understanding of the dynamic interplay between melanoma, immune checkpoints and immune cells, and to provide an update on currently investigated ICIs, beyond anti-PD-1 and anti-CTLA-4 agents.

**Abstract:**

More than ten years after the approval of ipilimumab, immune checkpoint inhibitors (ICIs) against PD-1 and CTLA-4 have been established as the most effective treatment for locally advanced or metastatic melanoma, achieving durable responses either as monotherapies or in combinatorial regimens. However, a considerable proportion of patients do not respond or experience early relapse, due to multiple parameters that contribute to melanoma resistance. The expression of other immune checkpoints beyond the PD-1 and CTLA-4 molecules remains a major mechanism of immune evasion. The recent approval of anti-LAG-3 ICI, relatlimab, in combination with nivolumab for metastatic disease, has capitalized on the extensive research in the field and has highlighted the potential for further improvement of melanoma prognosis by synergistically blocking additional immune targets with new ICI-doublets, antibody–drug conjugates, or other novel modalities. Herein, we provide a comprehensive overview of presently published immune checkpoint molecules, including LAG-3, TIGIT, TIM-3, VISTA, IDO1/IDO2/TDO, CD27/CD70, CD39/73, HVEM/BTLA/CD160 and B7-H3. Beginning from their immunomodulatory properties as co-inhibitory or co-stimulatory receptors, we present all therapeutic modalities targeting these molecules that have been tested in melanoma treatment either in preclinical or clinical settings. Better understanding of the checkpoint-mediated crosstalk between melanoma and immune effector cells is essential for generating more effective strategies with augmented immune response.

## 1. Introduction

Melanoma comprises a highly malignant cutaneous tumor with a rising incidence over the last decades. In the United States, melanoma incidence increased by more than 220% between 1975 and 2018, growing from 7.9/100.000 to 25.3/100.000, with an estimate of 43.7/100.000 by 2027 [1]. Although it accounts for less than 5% of skin cancers, melanoma is responsible for more than 75% of deaths attributable to cutaneous tumors [2], causing more than 57,000 deaths per year globally [3]. Despite these worrying numbers, mortality rates declined by 2% per year between 2016 and 2020 [1] due to the introduction of immune checkpoint inhibitors (ICIs) in the therapeutic algorithms of melanoma, with the overall survival (OS) rate for the metastatic disease reaching 41% at 5 years for pembrolizumab [4] and 49% after a median follow-up of 6.5 years for the nivolumab and ipilimumab combination [5]. From its generation under the effect of ultraviolet exposure, melanoma exhibits a relatively high mutational burden that is phenotypically expressed as tumor neoantigens [6]. These neoantigens are able to drive T-cell priming and activation, amplifying a tumor-specific immune response [7]. Arguably, under chronic tumor-antigen exposure and persistent TCR stimulation, melanoma-infiltrating CD8^+^T-cells can become dysfunctional and exhausted, allowing immune evasion and melanoma progression. However, the effects of factors including neoantigens, native-regulatory cytokines and immune suppressive cells (Tregs) in driving T-cell exhaustion still remain controversial, warranting further research [8,9]. These inducible exhaustion events can be effectively addressed with the administration of ICIs, which improve clinical outcomes either as monotherapies or, more recently, in combinatorial regimens. More insightful data on the underlying mechanisms of immunotherapeutic interventions on T-cell exhaustion imply that ICI and, particularly, anti-PD-1 therapy do not reverse the actual process of exhaustion but rather expand “stem-like” pools of proliferation-competent precursors of exhausted cells (Tpex), which are associated with enhanced tumor regression [10,11,12]. As of now, the Food and Drug Administration (FDA) has approved the anti-PD-1 agents nivolumab and pembrolizumab for adjuvant or metastatic settings; the anti-CTLA-4 agent ipilimumab (as monotherapy in anti-PD-1 refractory cases or in combination with nivolumab as a first-line treatment); and, more recently, the LAG-3 inhibitor, relatlimab, in combination with nivolumab in metastatic settings. Even with the approved ICIs, many patients show minimal benefit or early relapse, developing primary or acquired resistance to the ICI-enhanced immunosurveillance [13,14,15]. Among the studied resistance mechanisms, the expression of immune checkpoints beyond PD-1 and CTLA-4 is a well-documented phenomenon, supported by a volume of preclinical and clinical data [16]. The promising results of RELATIVITY-047 on the dual inhibition of PD-1 and LAG-3 opened up new horizons for combining two or potentially three ICIs in order to overcome this immune-mediated resistance mechanism. In the present study, we provide a comprehensive overview of novel checkpoint molecules including LAG-3, TIGIT, TIM-3, VISTA, IDO1/IDO2/TDO, CD27/CD70, CD39/73, HVEM/BTLA/CD160 and B7-H3 (Figure 1). Beginning from their immunomodulatory properties as co-inhibitory or co-stimulatory agents, we summarize here all the latest evidence from preclinical data to clinical-trial results on the immunotherapeutic agents that are currently investigated to target these molecules in the context of melanoma treatment.

## 2. LAG-3 (Lymphocyte Activation Gene-3, CD223)

LAG-3 protein is a cell-surface type-I transmembrane co-inhibitory receptor which binds to MHC class II, effectively blocking immune cells’ proliferation and functionality [17,18,19]. The encoding gene is localized distally on the short arm of chromosome 12, adjacent to the CD4 gene, and shares ~20% identical peptide sequences with the latter. LAG-3 is composed of 498 amino acids and features ectodomain homology with CD4, bearing four extracellular immunoglobulin superfamily (Ig)-like domains (D1-D4). Yet, LAG-3 leverages an additional 30 amino-acid loops in D1 to bind to MHC class II on APCs with higher affinity [17,18,19]. LAG-3 can also undergo metalloprotease-mediated cleavage between membrane-proximal D4 and the transmembrane region, to readily release a soluble form (sLAG-3), with shedding potentially restricting its inhibitory effects. Regarding the intracytoplasmic tail of LAG-3, it primarily engages in downstream signal transduction, while it encompasses three phylogenetically conserved motifs: a potentially phosphorylatable serine (Ser484); the KIEELE motif; and a unique, highly repetitive tandem glutamic acid–proline repeat (EP motif); however, data on their distinctive functional interplays remain ambiguous [20,21]. Recently published data, however, underscored that the EP motif of the cytoplasmic tail is intriguingly required to carry out LAG-3-mediated inhibition of proximal TCR signaling, independently of canonical ligand MHC II absence [22]. In the immunological synapse where LAG-3 molecules migrate and are constitutively and spatially associated with the TCR-CD3 complex, the EP motif effectively lowers the local IS pH, while it further provokes dissociation of the Src-family kinase Lck from CD4 and CD8 co-receptors, via sequestration of Zn^2+^, to disrupt T-cell activation signals [22]. Interestingly, the LAG-3/MHC II interaction is conformational-dependent and preferentially tethers to stable peptide-MHC II complexes (pMHC II), without outcompeting CD4 for binding. Therefore, elimination of the pMHC II-LAG-3 binding capacity can create anti-cancer immunity [23,24]. Galectin-3 (Gal-3) [25] fibrinogen-like protein 1 (FGL1) and liver sinusoidal endothelial cell lectin (LSECtin) comprise additional LAG-3 ligands [25,26]. An in-vitro study on B16 melanoma cells showed that LAG-3/LSECtin interaction can inhibit IFN-γ production and disrupt tumor-specific effector responses, which, in turn, may be reversed by LAG-3 blockade [27]. LAG-3 expression is induced on CD4^+^ and CD8^+^T-cells upon antigen stimulation, but not on naïve T cells. Consistent and robust LAG-3 upregulation due to sustained antigen exposure can result in functionally impaired T cells [28]. This role of LAG-3 in T-cell homeostasis seems to be directly or indirectly intertwined with Treg activity. Indeed, it is highly expressed on induced CD4^+^CD25^+^Tregs, at 20- to 50-fold, compared to effector/memory cells’ post-adoptive transfer. In-vitro assays have shown that anti-LAG-3 mAb can fully block the suppression of naive CD4^+^TCR transgenic T cells’ proliferation with CD4^+^LAG-3^+^suppressors [29]. These findings were confirmed in vivo, as an LAG-3 blockade curtailed Treg-mediated protection from lethal pneumonitis, revealing LAG-3’s role in conveying maximal suppressive activity to both induced and natural Tregs [29]. Camisaschi et al. concluded that LAG-3 is expanded in the TME of advanced melanoma and affects a distinct tumor-infiltrating CD4^+^CD25^high^FoxP3^+^Treg subpopulation which secretes immunosuppressive cytokines. These Tregs exert their suppressive effects selectively in tumor-invaded lymph nodes and lymphocytes infiltrating visceral or cutaneous metastases [30,31]. LAG-3^−/−^T-cells circumvented these Treg-induced effects and differentiated towards the Th1 phenotype.

On the other hand, LAG-3 expression sensitized T cells to Treg suppression by decreasing STAT5 signaling and promoting the FoxP3^+^Treg immunophenotype [29]. In melanoma, LAG-3/MHC-II interaction was shown to mediate TLR-independent induction of tolerogenic human pDCs in TME in vivo, maintaining immunosuppression [32]. Hemon et al. highlighted its impact on immune escape and cancer survival, as both sLAG-3 and LAG-3-transfected cells can protect MHC II-positive melanoma cells, but not MHC II-negative cells, from FAS-mediated and drug-induced apoptosis, through promotion of MAPK/Erk and PI3K/Akt survival pathways through the interaction of LAG-3 with MHC II was expressed on melanoma cells [33]. LAG-3, along with its ligands Galectin-3 and HLA-II, was found to be particularly expressed in melanoma lesions with inflamed T-cell phenotypes such as high-risk uveal melanoma, where Gal-3/LAG-3 interplay restricts T-cell-mediated responses [34]. Furthermore, high pre-treatment serum levels of sLAG-3 correlated with anti-PD-1 resistance (DCR: *p* = 0.009; PFS: *p* = 0.018), while increased infiltration with LAG3^+^ TILs in the TME of metastatic sites correlated with shorter PFS under PD-1 inhibition (*p* = 0.07) [35]. Taken together, LAG-3 co-targeting can be reasonably employed to augment anti-tumor immunity in combinational ICI treatment. In the case of a positive CIITA, a master regulator of MHC II, dual anti-LAG-3/PD-1 ICI attained profound antitumor impact and halted tumor growth in mice [36]. In an autologous TILs/melanoma co-culture system, concurrent anti-PD-1/LAG-3 treatment improves tumor control via numerically increasing the CD8^+^T-cells and significantly raising the T cell/melanoma cell ratio. In addition, this combination regimen was found to induce T cells’ responses since it elicited stronger IFN-γ production and more potent cytotoxicity exerted by CD8^+^T-cells [37]. In B16 melanoma tumors, dual LAG-3 and PD-1 depletion on TILs diminished tumor-induced tolerance and yielded higher responses compared to single-checkpoint-deficient mice, eliminating 80% of tumors (vs. 40% elimination in *PDCD1*^−/−^, and no tumor growth control in wild-type and *LAG-3*^−/−^mice). Peripheral blood profiling of melanoma patients who were candidates for immunotherapy showed that those with the LAG-3^+^immunotype had poorer outcomes after ICIs treatment compared to patients with the LAG-3^−^ immunotype (median OS: 22.2 vs. 75.8 months, respectively; *p* = 0.031), regardless of other immune-biomarker status (PD-1, TMB) [38].

Following the abovementioned data, LAG-3 was targeted in clinical testing using different agents: anti-LAG-3 mAbs, soluble LAG-3 immunoglobulin fusion proteins and anti-LAG-3 bispecific Abs (bsAbs) [39]. Table 1 presents all ongoing clinical trials testing LAG-3-targeted modalities in melanoma. Relatlimab (BMS-986016) was the first anti-LAG-3 mAb to receive FDA approval (March 2022), in combination with nivolumab (Opdualag™), for unresectable or metastatic melanoma for patients over 12 years of age based on the results of RELATIVITY-047 trial (NCT03470922). RELATIVITY-047 is a phase II/III, double-blind, randomized trial comparing relatlimab plus nivolumab to nivolumab alone in 714 untreated melanoma patients. After a median follow-up of 13.2 months, the ICI doublet significantly increased PFS compared to nivolumab monotherapy (10.1 months vs. 4.6 months, respectively; *p* = 0.006). This PFS benefit was obvious across all key subgroups. Opdualag also displayed a favorable safety profile, being associated with fewer serious treatment-related adverse effects (Grade 3/4 TRAEs in 18.9%) compared to the nivolumab–ipilimumab doublet (Grade 3/4 TRAEs in 55%) [40,41]. In the neoadjuvant setting for resectable melanoma (NCT02519322), the synergistic relatlimab/nivolumab combination induced a 57% pCR rate and 70% overall pathologic-response rate in the ITT population (*n* = 30), with no grade 3/4 TRAEs. The 1- and 2-year RFS rates were 100% and 92%, respectively, for patients with any pathologic response, as compared with 88% and 55% for patients without a pathologic response (*p* = 0.005) [42]. In addition to ICI-naïve disease, relatlimab is also being evaluated in anti-PD-1/L1 refractory melanoma (NCT01968109), where relatlimab plus nivolumab was well tolerated and showed desirable efficacy which corresponded with LAG-3 expression [43]. Currently, relatlimab is under investigation in more than 50 ongoing clinical trials, with many other anti-LAG-3 options also being tested. For instance, a phase-I study (NCT04640545) treated a population of 88 patients with advanced/metastatic melanoma with or without prior anti-PD-1 therapy, with LBL007 (fully human IgG4 anti-LAG-3 mAb) in combination with toripalimab (anti-PD-1 mAb). From preliminary data, out of the 32 radiologically evaluable patients, ORR was 12.5% and DCR was 53.1% [44]. Fianlimab (REGN3767) is another IgG4 fully human anti-LAG-3 mAb administered in combination with or without cemiplimab in a phase-I study of patients with advanced cancers (NCT03005782, *n* = 333). In a subgroup analysis of an advanced-melanoma cohort (*n* = 80, July 2022), fianlimab plus cemiplimab demonstrated encouraging activity across subgroups with unfavorable features at baseline, such as liver metastases, where ORR was 47.4%, DCR was 63.2%, and mDOR was 9.0 months [45]. Phase-III trials investigating the superiority of fianlimab plus cemiplimab over pembrolizumab in untreated advanced melanoma (NCT05352672) or as adjuvant therapy in completely resected high-risk melanoma (NCT05608291) are ongoing. Similarly, another randomized phase-II study (NCT03484923) examined the capability of combinations of the anti-PD-1 mAb spartalizumab to counteract the resistance in melanoma that progressed under PD-1 blockade. The addition of ieramilimab (LAG525) to spartalizumab was found to be more likely to induce responses in LAG-3^+^metastatic melanoma [46]. Ieramilimab is a humanized IgG4 anti-LAG-3 mAb found to block the LAG-3/FGL-1 and LAG-3/MHC II interactions. In a phase-I/II clinical trial (NCT02460224, *n* = 255 as of cutoff) testing ieramilimab as a monotherapy or combined with spartalizumab in advanced tumors including melanoma, a moderate anti-tumor effect was observed in the combination arm, while toxicities were manageable [47]. In a different LAG-3 targeting proof-of-concept, eftilagimod alpha (IMP321) is a first-in-class soluble LAG-3 Ig fusion molecule (hLAG-3Ig), which has been used as an immunostimulatory agent for vaccines and has showed durable, tumor-specific T-cell responses in metastatic melanoma patients, independently of HLA class-II genotypes [48]. More specifically, in a phase-I study (NCT02676869), when administered in combination with pembrolizumab, IMP321 displayed sustained anti-tumor activity with no DLTs or AEs ≥ grade 4. This strategy potentiated long-term increases in circulating activated CD8^+^and CD4^+^T-cell counts, while promoting IFN-γ and Th1-mediated cytokines [49]. Lately, newer bsAbs can selectively identify and bind concurrently with two different checkpoint receptors. For instance, the bsAb-mediated LAG-3xPD-1/-L1 blockade had encouraging preclinical results [50,51,52] and RO7247669, an anti-PD-1xLAG-3 bsAb, is currently being tested in clinical trials (NCT04140500, NCT05419388, NCT05116202) for advanced/metastatic tumors, including melanoma. In its first-in-human phase-I study (NCT04140500), this bsAb showed an ORR of 17.1%, and DCR of 51.4% in ICI-pretreated patients [53]; NCT05419388 and NCT05116202 have no available results yet, with the latter being an umbrella study of agents in treatment-naive, resectable stage-III or pretreated stage-IV melanoma [54].

## 3. TIGIT (T-Cell Immunoglobulin and ITIM Domain)

TIGIT is another co-inhibitory molecule which belongs to the poliovirus receptor/nectin family of proteins, a subset of the immunoglobulin (Ig) superfamily. It is composed of an extracellular Ig variable domain, a type-I transmembrane domain, and a cytoplasmic tail with two inhibitory motifs: an ITIM and an Ig tail-tyrosine-like motif [55,56,57]. TIGIT has been found to be highly expressed in CD4^+^, CD8^+^T-cells, Tregs and NK cells, while it is absent or weakly expressed in naïve T-cell populations [55,58,59,60]. CD155 is the main ligand of TIGIT, while CD112 and CD113 have lower affinity. CD155, CD112 and CD113 ligands belong to the nectin family [55,58,61]. TIGIT exerts co-inhibitory properties binding CD155 with higher affinity than DNAM-1 (CD226), which is the co-stimulatory receptor, a binding process which is similar to that between CTLA-4 and CD28 [55,62,63,64]. Both CD155 and CD112 have been observed to be overexpressed in various human malignancies [65,66,67,68]. TIGIT has been shown to be upregulated in melanoma cells [69,70], which negatively affect tumor-specific CD8^+^T-cells via CD155 interaction [71]. In particular, TIGIT restricts TCR-induced p-ERK signaling in CD8^+^T-cells [72] and inhibits their activation and proliferation [73]. Upon binding to CD155, TIGIT also suppresses NK-mediated cytotoxicity and IFN-γ production [58,74], while blocking TIGIT/CD155 interaction effectively reverses NK cells’ dysfunction and confers anti-tumor activity, even in melanoma models refractory to melanoma-associated antigen specific CD8^+^T-cells [75,76]. To this end, reduced NK cell activity has been correlated with higher TIGIT expression levels [77]. TIGIT expressed by Tregs alters cytokine homeostasis and inhibits proinflammatory Th1 and Th17 cell responses, while it does not affect Th2-mediated responses [78,79]. In melanoma patients, an increased TIGIT/CD226 ratio in tumor-residing Tregs (increased TIGIT expression and decreased expression of its competing costimulatory receptor CD226) is associated with highly suppressive Treg function and poor clinical outcomes upon ICI blockade [80], while circulating populations of PD-1^+^TIGIT^+^CD8^+^T-cells can predict the efficacy of anti-PD-1 therapy [81]. Similar results were reported by Lee et al., as high expression of either LAG-3 or TIGIT on TILs was associated with worse survival [82]. These results provide the rationale for co-inhibiting TIGIT together with PD-1 and/or LAG-3; for the monitoring of circulating PD-1^+^TIGIT^+^CD8^+^T-subsets or of LAG-3^+^TIGIT^+^TILs on melanoma biopsies as early unfavorable cellular biomarkers of an anti-PD-1 response; and for activating CD226 in Tregs together with TIGIT blockade, to counteract Treg suppression in melanoma patients.

TIGIT is currently being targeted as part of combinatorial regimens in many ongoing clinical trials (Table 2). Sub-studies 02A, 02B, 02C of the KEYMAKER-UO2, which are in phase I/II, are currently recruiting patients with advanced melanoma in order to test the efficacy and safety of different ICI-based combinations. The 02A sub-study (NCT04305041) randomizes patients with PD-1 refractory melanoma to receive vibostolimab (MK-7684) (anti-TIGIT), pembrolizumab and quavonlimab (anti-CTLA-4); pembrolizumab, quavonlimab and lenvatinib; or pembrolizumab and ATRA. In the 02B sub-study (NCT04305054), patients with previously untreated advanced melanoma will receive pembrolizumab and vibostolimab, pembrolizumab alone or pembrolizumab in combination with other agents. In the 02C sub-study (NCT04303169), participants with stage-III melanoma who are candidates for neoadjuvant therapy will receive vibostolimab and pembrolizumab, pembrolizumab alone or pembrolizumab in combination with other agents. In the adjuvant setting, a currently recruiting phase-III trial (NCT05665595) is being conducted to compare pembrolizumab and vibostolimab with pembrolizumab alone for resected high-risk melanoma. RFS was set as its primary outcome and an estimated enrollment of up to 1560 participants is anticipated. Another phase-II single-arm study is currently recruiting participants with PD-1 relapsed/refractory melanoma to receive the anti-TIGIT domvanalimab (AB154) in combination with the anti-PD-1 zimberelimab (AB122) to assess the response of this combination (NCT05130177). Tiragolumab (RG6058) is another TIGIT inhibitor and is currently being used in four phase-II melanoma clinical trials in combinatorial regimens (NCT05483400, NCT05116202, NCT05060003, NCT03554083). The most recent TIGIT inhibitor, EOS-448, is being examined in combination with other agents in patients with melanoma and other solid tumors to evaluate its safety and efficacy (NCT05060432).

## 4. TIM-3 (T-Cell Immunoglobulin- and Mucin-Domain-Containing Molecule-3)

TIM-3 comprises a transmembrane inhibitory protein that contains an amino-terminal Ig variable domain (V domain) with five noncanonical cysteines, a mucin stalk, a transmembrane domain and an inhibitory cytoplasmic tail. The main identified TIM-3 ligands include Galectin-9 [83], CEACAM1 [84], HMGB1 and phosphatidylserine [85], which bind to distinct regions of the extracellular Ig V domain. TIM-3 is mainly expressed on CD4^+^T, CD8^+^T-cells, Tregs as well as on innate immunity cells [86]. In cancer, TIM-3 and its cognate ligands were shown to be markedly upregulated, forming a suppressive TME for tumor-residing immune cells. In melanoma, expression of TIM-3 on TILs restricts secretion of proinflammatory cytokines, inhibits Th1-mediated responses and is correlated with more severe T-cell exhaustion compared to TIM-3^−^PD1^+^CD8^+^T-cells [87]. Moreover, TIM-3 expression is correlated with NK cell dysfunction, resulting in reduced secretion of IFN-γ and decreased cytotoxic activity [88]. TIM-3 expression in Tregs enhances their suppressive capacity and is potentially STAT-3-induced, as STAT-3 inhibition in anti-PD-1 refractory tumors leads to treatment response via TIM-3 downregulation on Tregs [89]. Regarding TIM-3 ligands, binding of HMGB1 dysregulates DCs’ activation and function [90], while CAECAM1 was found to be upregulated in metastatic specimens from melanoma patients with disease progression under PD-1 inhibition [91]. Galectin-9 simultaneously interacts with both TIM-3/PD-1 and TIM-3/VISTA to promote apoptosis of TIM-3^+^CD8^+^T-cells [87]. Importantly, TIM-3 upregulation has been shown upon MEK inhibition with trametinib in melanoma patients, resulting in CD8^+^T-cell deprivation, arguing for an adaptive resistance mechanism [92]. Many studies have pinpointed TIM-3 as an exhaustion and prognosis biomarker in melanoma. Tallerico et al. studied 67 melanoma patients treated with ipilimumab and correlated higher PD-1 and TIM-3 expression with poorer clinical outcomes [93]. Out of 16 metastatic melanoma patients receiving pembrolizumab monotherapy, Graves et al. found that non-responders had higher TIM-3 expression on the surface of CD8^+^T-cells (*p* = 0.047) [94]. Machiraju et al. observed that in metastatic melanoma, decreased PFS under anti-PD-1 treatment is associated with a higher amount of LAG3^+^ (*p =* 0.07) and TIM-3^+^ (*p *= 0.019) TILs [35]. Higher TIM-3 expression on CD8^+^T-cells was also associated with decreased RFS (*p =* 0.074) and increased tumor volume (*p* = 0.0472) in melanoma subjects under ipilimumab plus IFNa [95]. Recently, Conway et al. found lower T-cell infiltration in the TME of melanoma metastases with poorer prognosis (liver *p *= 0.0116 and brain *p* = 0.0252). In comparison to other organ metastases, lower PD-1 and higher TIM-3 expression were displayed in liver metastases (liver *p* < 0.05, lymph node *p* = 0.0004, sub-cut *p* = 0.0082 and brain *p* = 0.0128) [96].

Selective inhibition of either TIM-3 or its ligands has produced notable preclinical anti-melanoma efficacy. In mouse models, a dual PD-1 and TIM-3 blockade reversed T-cell exhaustion and was more efficacious than anti-PD-1 monotherapy (median OS 11.9 weeks vs. 5 weeks, respectively (*p *= 0.0008)) [97]. This strategy is currently being studied in several clinical trials, with promising early results (Table 3). TSR-022 (Cobolimab) is an anti-TIM-3 mAb studied in three such studies (NCT04139902, NCT02817633). In the phase-I AMBER study (NCT02817633), cobolimab is being tested as a monotherapy or in combination with nivolumab or dostarlimab (anti-PD-1) in patients with advanced solid malignancies; preliminary evidence shows anti-tumor activity and a manageable safety profile [98]. The randomized phase-II study NCT04139902 compares cobolimab plus dostarlimab to dostarlimab alone in patients with operable, regionally advanced or oligometastatic melanoma. Sabatolimab (MBG453) is another anti-TIM-3 agent that blocks TIM-3 interaction with its ligands Galectin-9 and PtdSer. In preclinical models, sabatolimab showed potent antitumoral activity via enhanced CD8^+^T-cell killing activity, increased DC-mediated cytokine production and increased the ADCC of TIM-3^+^targeted cells [99]. In a phase-II study, sabatolimab plus spartalizumab has a favorable safety profile; however, it exhibited limited efficacy in pretreated melanoma patients. INCAGN02390 comprises a fully human Fc-engineered IgG1κ anti-TIM-3 mAb that prevents PtdSer interaction and was shown to be well-tolerated in patients with advanced malignancies according to a phase-I trial (NCT03652077) [100]. Different combinations of INCAGN02390 with PD-1 and LAG-3 blocking agents are being evaluated in a phase-I/II study (NCT04370704) for patients with advanced cancers, including melanoma. Lastly, a phase-I trial (NCT03708328) examines single-agent RO7121661, a bsAb binding to both TIM-3 and PD-1, in patients with different types of cancers, including melanoma, while NCT05451407 is scheduled to test the efficacy and safety of TQB2618 injection (humanized IgG4 mAb targeting TIM-3) together with terprizumab (anti-PD-1) in 50 patients with advanced melanoma.

## 5. VISTA (V-Domain Immunoglobulin Suppressor of T-Cell Activation)

VISTA is a recently described multi-lineage IC with a pleiotropic immunosuppressive effect that stems from its receptor/ligand bifunctionality [101,102]. VISTA comprises a 279 amino-acid-long type-I transmembrane protein consisting of a single Ig-V domain linked to a stalk structure, a transmembrane domain and a cytoplasmic tail that is encoded by the Vsir gene [103,104]. Primary amino-acid-sequence analysis has aligned VISTA with the CD28/B7 family, with PD-L1 being the closest related protein, exhibiting a modest 24% sequence identity [105]. Although, VISTA’s ligands are yet to be conclusively documented, V-set and immunoglobulin domain containing 3 (VSIG3) [106] along with Galectin-9 [107] and MMP-13 [108] have been described as physiologically significant ligands. Intercellular VISTA homophilic interactions have also been described [109]. In the non-malignant setting, VISTA has a broad expression in hematopoietic tissues, with the highest presentation in CD11b^+^subpopulations [102]. It is pronouncedly expressed in naïve CD4^+^T-cells and FoxP3^+^ Tregs, less present on CD8^+^, while NK cells are generally VISTA^−^ [102]. B-cells display non-detectable [102,110] or low [105,111] VISTA expression. Plasma cells have been recently reported to be VISTA^+^ [112]. A body of evidence supports its immunosuppressive activities. VISTA activation on macrophages induces upregulation of IL-10 and IL-1Ra along with downregulation of IL-12, generating an M2-like immunosuppressive phenotype [113]. In the presence of high-dose lipopolysaccharides (LPS), an anti-VISTA agonistic mAb epigenetically reprogrammed macrophages and induced a tolerogenic cytokine profile [114]. In accordance with these findings, Vsir knocked-out mice showed enhanced TLR-mediated proinflammatory cytokine production [104] and increased numbers of activated T-cells, indicative of the loss of peripheral cellular homeostasis [101,115]. In activated T-cells, VISTA/VSIG3 interaction has been found to inhibit T-cell proliferation along with significant reduction in proinflammatory cytokines and chemokines [104]. Anti-VISTA mAbs effectively attenuated these effects [106]. Galectin-9, the second reported VISTA ligand, was originally described as a TIM-3 ligand that induces apoptosis of TIM-3^+^CD8^+^T-cells [116]. However, Yasinska et al. recently reported that Galectin-9 also has high affinity with VISTA, resulting in TIM-3/VISTA/Galectin-9 cell-surface complexes that trigger programmed cell death through granzyme B accumulation and other proapoptotic pathways [107]. The clinical significance of VISTA’s binding on T-cells was first documented by Wang et al. who reported that a VISTA-Ig fusion protein inhibited CD4^+^ and CD8^+^T-cell proliferation and cytokine production [105] via induction of FoxP3^+^Tregs from naïve CD4^+^T-cells [117]. More recently, VISTA was recognized as an acidic-pH selective ligand for the co-inhibitory receptor P-selectin glycoprotein ligand-1 (PSGL-1) [118], which is upregulated in exhausted T-cells [119].

In melanoma, VISTA expression in tumor-infiltrating immune cells in 85 primary melanoma specimens was associated with significantly worse disease-specific survival (*p* = 0.005) and emerged as an independent negative prognostic factor in multivariate analyses (*p* = 0.02) [120]. Furthermore, Choi et al. documented that VISTA is highly expressed in 27% of melanoma patients (37/136) and that high VISTA expression is correlated with advanced disease stage (*p* = 0.008) and inferior median OS (58.0 vs. 79.0 months, *p* = 0.017) [121]. Except for tumor-related immune cells, Rosenbaum et al. reported that VISTA is also present in melanoma cells, identifying low or moderate and high VISTA expression in melanoma cells of 12 of 13 patient samples. In the same study, in VISTA^+^mouse melanoma cell lines, increased numbers of FoxP3^+^Tregs (*p* < 0.05) and decreased MHC-II levels un dendritic cells (*p* < 0.05) were observed compared to VISTA^−^models [122]. Furthermore, in melanoma patients that progressed after initial response to anti-PD-1 monotherapy or anti-PD-1 plus ipilimumab, the density of VISTA^+^lymphocytes was significantly increased (*p* = 0.009) compared to matched pre-treatment levels, arguing for an adaptive resistance mechanism [123]. Similar results were provided by the study of Gao et al., where ipilimumab treatment resulted in significantly higher VISTA^+^CD4^+^T-cells (*p* = 0.001), CD8^+^T-cells (*p* = 0.03) and CD68^+^macrophages (*p* = 0.03) compared to pretreatment levels [124].

Based on these data, anti-VISTA mAbs have been studied in preclinical studies and recently in clinical-trial settings (Table 4). Le Mercier et al. presented that anti-VISTA mAb administration restricts the emergence and the suppressive potential of Tregs, increases the infiltration and effector function of CD8^+^T-cells in the TME, and suppresses the growth of melanoma in mice [117]. CA-170 is a first in class, oral PD-L1/L2 and VISTA dual inhibitor that has shown significant anti-tumor properties in different mouse cancer models, including metastatic melanoma [125,126]. CA-170 has completed a phase-I trial for the treatment of patients with lymphoma or advanced solid cancers, including melanoma, (NCT02812875) showing acceptable safety and increased numbers of circulating activated CD4^+^and CD8^+^T-cells [125]. Early phase-II results from the non-small-cell lung carcinoma subgroup showed potent clinical activity [127]. HMBD-002 is an Fc-independent anti-VISTA mAb that has been shown to prevent VISTA interaction with VSIG3, but not with PSGL-1, as the histidine residues implicated in the VISTA/PSGL-1 interaction are distal to the HMBD-002 epitope [128]. HMBD-002 was shown to restrict MDSC-induced T-cell suppression, neutrophil chemotaxis and promote a Th1 immune response. HMBD-002 is currently being tested in a phase-I trial as a monotherapy or in combination with pembrolizumab in patients with advanced solid tumors (NCT05082610). CI-8993 comprises another anti-VISTA mAb currently being tested in a phase-I clinical trial for advanced solid malignancies (NCT04475523). SG7 is another published anti-VISTA mAb that prevents VISTA interaction with both VSIG3 and PSGL-1 and significantly inhibits tumor growth in mouse melanoma lines [129]. Lastly, metformin was shown to inhibit VISTA and restrain growth rate in melanoma cells in vitro and in vivo [130].

## 6. IDO1 (Indoleamine 2,3-Dioxygenase 1), IDO2 (Indoleamine 2,3-Dioxygenase 2), TDO (Tryptophan 2,3-Dioxygenase)

IDO1, IDO2 and TDO are immunoregulatory enzymes that have emerged as metabolic checkpoint molecules as they are implicated in suppressing T-cell immune responses via the conversion of tryptophan into kynurenines. In noncancerous conditions, TDO is constitutively expressed in the liver with minimal expression in other normal tissues [131]. IDO is present in the epithelial and endothelial cells of several organs, while it is also expressed in immune cells, primarily of the myeloid lineage, under stimulation by proinflammatory cytokines [131,132]. Contrary to a steady state, markedly elevated TDO and IDO levels have been observed in multiple human tumors, including melanoma [133,134]. T-lymphocytes are extremely sensitive to tryptophan shortage, which causes cell-cycle arrest and enhanced susceptibility to Fas-mediated apoptosis [16,135]. On the other hand, tryptophan accumulation downregulated PD-1 on the surface of CD8^+^T-cells and potentiated their cell-killing effect on co-cultured melanoma cells [136]. Contrary to tryptophane, kynurine abrogates CD8^+^T-cell cytotoxicity and viability [137,138]. In an IDO-high TME, infiltrating CD8^+^T-cells were substantially reduced, while the remaining viable cells lost their capacity to lyse tumor cells due to the inhibition of cytokine and granule cytotoxic protein production [137]. In addition, IDO expression in human melanoma is positively correlated with the number of tumor-infiltrating MDSCs and Tregs, which further hamper CD8^+^T-cell effector functions [139]. Importantly, IDO upregulation in the TME was shown to be dependent on CD8^+^T-cell infiltration and IFN-γ stimulation, supporting an adaptive mechanism of resistance [140]. In line with these findings, a more than 50% increase in the kynurenine/tryptophan ratio compared to baseline was associated with worse survival in melanoma patients receiving nivolumab (mOS: 15.7 vs. 37 months, *p* = 0.00006) [141].

Based on these preclinical findings, a phase-III ECHO-301/KEYNOTE-252 trial (NCT02752074) randomized 706 patients with metastatic melanoma to receive either epacadostat, an IDO1 selective inhibitor, plus pembrolizumab or pembrolizumab plus placebo (Table 4). However, the experimental combination did not improve outcomes (mPFS: 4.7 vs. 4.9 months, mOS: not reached in either group) compared to the control arm [142]. The low dose of epacadostat (100 mg twice per day) in this trial was blamed for the negative results, as it was not considered sufficient to reach a maximum pharmacodynamic effect [143]. The compensatory activity of IDO2 and TDO could be another potential mechanism of resistance to single IDO1 inhibition [144], suggesting a total inhibition of IDO1/IDO2/TDO as a more effective strategy to manipulate the tryptophan–kynurenine pathway [145]. In a phase-I/II trial, a more potent IDO1 inhibitor, BMS-986205, was added to nivolumab or nivolumab plus ipilimumab treating 627 patients with advanced cancers including melanoma, and results are awaited (NCT02658890). Contrary to epacadostat, indoximod, a tryptophan mimetic, demonstrated promising efficacy when combined with pembrolizumab, nivolumab or ipilimumab (mPFS = 12.4 months, ORR = 51%, CR = 20%) in a phase-II trial (NCT02073123) [146]. IO102/IO103, a first-in-class immunomodulatory vaccine against IDO and PD-L1, plus nivolumab was tested in a phase-I/II trial (NCT03047928) and produced notable efficacy (ORR = 80%, CR = 43%, mPFS = 26 months) along with comparable efficacy to nivolumab monotherapy [147]. In preclinical models, a dual inhibitor of topoisomerase II and IDO1 [148]—a drug conjugate of D-1MT, an IDO inhibitor and paclitaxel [149], and a drug conjugate of D-1MT and gallic acid (GA-1MT) [150]—were described to have potent antiproliferative activity against melanoma cells.

## 7. CD27/CD70

The CD27/CD70 axis comprises a co-stimulatory ligand–receptor pair that belongs to the TNF superfamily and their interplay is involved in both innate and adaptive immunity responses [151]. CD27 is a transmembrane type-I glycoprotein receptor largely expressed in resting CD4^+^and CD8^+^T-cells, Tregs [152], B-cells [153], NK populations and hematopoietic progenitors [154]. The CD70 ligand constitutes a type-II transmembrane glycoprotein that is upregulated upon antigen challenge in antigen-activated B-and T-cells, NK cells and APCs [154]. Notably, CD27 and CD70 surface expression patterns are reciprocally influenced and inversely correlated, as CD27 engagement subsequently downregulates CD70 [155]. Upon interaction with its ligand on APCs and lymphocytes, CD27 modulates the expansion of effector and memory T-cell repertoires by promoting T-cell priming and survival [156]. During immune response, mature-skin Langerhans cells display greater CD70 expression levels, which enable superior antigen-specific cytokine production and robust CD8^+^T-cell activation [157]. Fifteen years ago, Keller et al. noted that the expression of CD70 by otherwise-immature cDCs sufficed to convert CD8^+^T-cell tolerance into immunity, defining the importance of CD27–CD70 interactions at the interface between T-cells and DC [158]. Moving one step forward, adoptively transferred CD70-expressing immature DCs could prime CD8^+^T-cells, by CD27, to become tumor-eradicating cytolytic effectors and memory cells with a capacity for robust secondary expansion, independently of CD4^+^T-helpers. These data highlighted the importance of CD27/CD70 interactions at the T-cell/DC interface, indicating the crucial role of CD70 in the design of DC vaccination strategies [159].

On a different note, the tight regulation that governs co-stimulatory CD70/CD27 signaling can be aberrantly altered to have tumor-favoring consequences. In a pan-cancer analysis, CD70/CD27 signaling was, surprisingly, found to contribute to immune evasion, potentially through CD70-mediated Treg recruitment. CD70 was found to be upregulated across many cancers, while CD70/CD27 expression correlated with Treg infiltration levels [160]. Ιn the malignant setting, a CD27-mediated signal can, strikingly, overrule anti-tumor immunity. Indeed, CD70/CD27 signaling has been associated with maintained Treg survival, function and intratumoral accumulation [161]. Accordingly, steady-state DCs remain tolerogenic due to Treg-expressed CD27 which impedes CTL priming and favors tumor outgrowth [162]. Ectopic, melanoma-expressed CD70 is implicated in melanoma invasiveness, with monomeric CD70 diminishing metastatic potential in vitro and in vivo, whereas the trimeric form, as in mAb-mediated cross-linking, restores melanoma-cell migratory capacity [163]. Intriguingly, continuous engagement of CD27 by its ligand can result in proliferative exhaustion and CD27 signaling-induced T-cell apoptosis, potentially eliminating activated T-cells and compromising memory development [164]. On the other hand, in-vivo CD27 engagement by CD70 on B-cells has shown to augment primary CD8^+^T-cell responses and tumor rejection even in poorly immunogenic tumor models, as CD27 stimulation can boost antigen-specific CD8^+^T-cell expansion and concomitantly improve per-cell cytotoxic functionality [165]. Studies on CD27 and CD70 expression levels on bulk TILs administered to melanoma patients after chemotherapy indicated that CD27^+^CD8^+^and CD70^+^CD8^+^TIL pools were significantly higher (*p* = 0.004 and *p* = 0.01, respectively) in responders and associated with tumor regression following adoptive cell transfer [166]. Arguing in favor of CD27 agonism, CD27 engagement with an agonist mAb downregulated co-inhibitory receptors in vitro, while it led to CTL-mediated tumor rejection in vivo [167]. In a murine melanoma model, anti-CD27 agonistic mAb treatment potentiated tumor control by increasing the frequency and maximal activity of tumor-specific CD8^+^T-cell and NK cells in the TME. Favorably, a ~9-fold increase in antigen-specific CD8^+^T-cells/FoxP3^+^CD4^+^Tregs ratio was observed and PD-1 levels on TILs were found to be attenuated [168].

The therapeutic potential of CD27 targeting can be exemplified with varlilumab, a first-in-class mAb that conferred CD27 activation in early studies (Table 4). Marked pro-inflammatory chemokine signatures corresponded with varlilumab-mediated stimulation, which, in turn, exerted indirect CD8^+^T-cell dominant activation, without Treg induction, in a TCR engagement-dependent manner [169]. Similarly, in hCD27 transgenic mice, varlilumab selectively strengthened CD8+ T-cell phenotypes in terms of activation, proliferation and cytotoxic state in lymphoid organs and tumor infiltrates, led to preferential depletion of suppressive Tregs, and generated durable and broad anti-tumor immunity [164]. In agreement with previous animal data, varlilumab’s first-in-human study (NCT01460134), which included a melanoma cohort (*n* = 16) among other solid tumors, showed a substantial reduction in circulating Tregs, transient elevation of serum chemokine levels and induction of activation status, favoring the relative ratio of terminally differentiated effector memory T cells over naïve cells. It thereby provided clinical evidence of the biological activity of varlilumab in advanced treatment-refractory tumors, while establishing an acceptable safety profile [170]. In a preliminary vaccine study, anti-CD27 agonist combined with PD-1 blockade re-enforced CD4^+^T-cell help and optimized CTL responsiveness for effective tumor elimination [171]. Anti-CD27 and PD-1/L1 blockade synergy has been described to reverse CD8^+^T-cell exhaustion and abrogate molecularly installed quiescence, optimizing effector differentiation and CD8^+^T-cell proliferation compared to either therapy alone [172,173]. In a transcriptome analysis, the CD27-activated effector phenotype was consistent with the promotion of a Myc-regulated gene-expression program and indicated upregulation of IL-2 signaling as a major output of CD27 stimulation. In this context, it reinvigorated endogenous anti-tumor CD8^+^T-cell responses and enhanced adoptive T-cell therapy in melanoma models, securing improved long-term survival and tumor protection versus monotherapy [173]. In a phase-I/II study (NCT02335918), varlilumab was safely administered in conjunction with nivolumab to patients with checkpoint-naïve chemotherapy-refractory solid tumors, including melanoma (Ph I, *n* = 4), to generate a strong immunologic response of proinflammatory nature with significantly decreased Tregs (Ph I; median decrease of 51%; *p* < 0.001, Ph II week 8; median decrease of 71%; *p* < 0.001) and predominantly CD8^+^ increased TILs. However, clinical outcomes were modest in the phase-I stage, with ORR reaching 6.7% and DCR 33.3% for the maximal dosage of varlilumab evaluated [174,175]. The exogenous anti-CD27 stimulation has demonstrated preferential adjuvant effects for peptide vaccines containing linked class-I/II epitopes and was more effective in amplifying antigen-specific CD8^+^T-cell responses in a CD4^+^helper-dependent manner, displaying anti-tumor efficacy in intracranial melanoma models [176]. A trial investigating a peptide vaccine (6 MHP) with adjuvants including varlilumab in stage-III–IV melanoma is currently ongoing (NCT03617328). Recent upgrades to the anti-CD27 mAbs include hexavalent; natural ligand-mimicking agonist HERA-CD27L, which has showed improved activity in vitro and in vivo compared to the bivalent clinical benchmark mAb [177]; as well as CDX-527, a tetravalent anti-CD27xPD-1 bsAb that outperformed the parental mAbs and their combination [178] and has now entered clinical investigation for advanced malignancies (NCT04440943). Lastly, several CD70-targeting agents have been investigated in CD70-expressing tumor indications and hematological malignancies, including anti-CD70 mAbs, ADCs and CAR T-cell therapy [154,179].

## 8. CD39 (Ectonucleoside Triphosphate Diphosphohydrolase-1, E-NTPDase1) and CD73 (Ecto-5′-Nucleotidase, Ecto5′NTase)

CD39 and CD73 are ecto-nucleosides that catalyze the dephosphorylation of ATP to produce adenosine [180]. At steady state, CD39 and CD73 are constitutively expressed in various cell types within the TME, including tumor cells, stromal cells, endothelial cells, and the infiltrating immune cells (e.g., B-cells, NK cells, DCs, macrophages and Tregs) [181]. Under hypoxic conditions, hypoxia-inducible factor 1a stimulates upregulation of CD39, CD73, adenosine 2A receptor (A2AR) and adenosine 2B receptor (A2BR) [182,183,184]. In melanoma lesions, CD39 and CD73 are upregulated on Tregs as well as on activated CD4^+^ and CD8^+^T-cells and exosomes. Accumulated adenosine, through its continuous binding on A2A receptors, impairs effector functions, such as cytokine secretion, cytotoxic activity and the metabolic fitness of activated CD8^+^TILs [185]. Further A2AR signaling also suppresses CD8^+^T-cell expression of CCR7, maintaining an anti-apoptotic role via the PI3K/AKT pathway [186], and induces T-cell apoptosis [187]. Except for the direct inhibitory activity of adenosine on CD8^+^T-cells, A2AR stimulation also confers an indirect suppressive effect through expansion of Tregs. It was shown that the coculture of Tregs with an A2AR agonist significantly increased the Treg count as well as CTLA-4 expression [188]. Activation of A2AR on NK cells in the TME restricts their proliferation, maturation [189], cytokine secretion and cell-killing activity [190,191]. A2BR and A2AR stimulation promotes intratumoral accumulation of MDSCs [192] while it delays DC activation and induces a tolerogenic phenotype with limited CD8^+^T-cell priming capacity [193,194]. In melanoma, high CD73 expression was associated with fewer intratumoral CD8^+^T-cells and NK cells (*p* < 0.05), while it also correlated with lower OS and DFS post-resection (*p* < 0.005 for both) [195]. In specimens from melanoma patients, CD39^+^CD8^+^T-cells exhibited an exhausted phenotype with limited secretion of proinflammatory cytokines and elevated expression of co-inhibitory receptors [196]. Furthermore, high pretreatment soluble CD73 was significantly associated with poor response to nivolumab monotherapy (mPFS = 2.6 vs. 14.2 months, *p* = 0.001) [197]. In a similar study, pretreatment CD73 was significantly lower in patients achieving OR with nivolumab and pembrolizumab alone or in combination with ipilimumab and emerged as an independent prognostic factor for both OS (*p* = 0.009) and PFS (*p* = 0.001) [198]. A low pretreatment number of circulating CD8^+^PD-1^+^CD73^+^T-cells was also associated with better OS under nivolumab monotherapy (22.4 vs. 6.9 months, *p* = 0.001) [199]. CD73 has been reported to be further upregulated in melanoma lesions progressing under anti-PD-1 treatment, arguing for an adaptive resistance mechanism [200,201]. Adenosine-mediated CAR T-cell suppression was also recently described [202].

Based on these data, the CD39/CD73/adenosine axis has emerged as a promising therapeutic target in melanoma (Table 4). Briceno et al. demonstrated that CD73^−^CD8^+^T-cells expressed lower levels of exhaustion markers and were more effective in reducing tumor burden in melanoma bearing mice, compared to CD73^+^CD8^+^T-cells [203]. An A2AR antagonist (Ciforadenant, CPI-441) was tested in 502 patients with solid cancers including melanoma as a monotherapy or in combination with atezolizumab in a phase-I study (NCT02655822). Although no results for the melanoma cohort have been published to date, preliminary data yielded no grade 3/4 TRAEs [204], while encouraging clinical activity was reported for prostate- [205] and renal-cancer patients [206]. In preclinical models, CPI-441 has demonstrated potent antitumor activity as it enhanced CD8^+^T-cell activation and potentiated the effects elicited by anti-PD-1 and anti-CTLA-4 therapy [207]. Further studies have published similar results upon co-administration of A2AR antagonists and anti-PD-1 anti-CTLA-4 mAbs melanoma mouse models [208,209]. Inupadenant (EOS-850) is another A2AR antagonist currently under investigation in a phase-I trial in patients with solid tumors refractory to a median of three prior standard regimens (NCT03873883). Although treatment was discontinued in seven subjects due to unacceptable toxicity, two PRs were reported in a preliminary report, one of which was in a melanoma patient [210]. Etrumadenant is a dual A2AR/A2BR antagonist that recently completed a phase-I trial (NCT03629756) in combination with zimberelimab (anti-PD-1), demonstrating a favorable safety profile [211]. In murine models, etrumadenant effectively augmented CAR T-cell activation and abrogated adenosine-induced suppression [202].

## 9. HVEM/BTLA/CD160

Engagement between a herpesvirus entry mediator (HVEM, TNFRSF14), B- and T-lymphocyte attenuator (BTLA) and CD160 comprises an immune checkpoint network that bridges two functionally distinct receptor superfamilies: the tumor necrosis factor receptor (TNFR) superfamily and the Ig superfamily [212]. HVEM is broadly found in T-cells, B-cells, NK cells and DCs and it is described as a bidirectional switch that delivers co-inhibitory or co-stimulatory signals depending on the cognate interactor [213]. The main ligands of HVEM include LIGHT and Lymphotoxin-A, which promote stimulatory signaling, and CD160 and BTLA which confer inhibitory effects [214]. Importantly, the latter two are competitive ligands of the CRD1 domain indicating that HVEM can receive both inhibitory and stimulatory signals at distinct binding sites [215]. CD160 is a transmembrane protein and is primarily expressed in NK cells, CD4^+^and CD8^+^T-cells [216]. Of the three CD160 forms, the GPI isoform (CD160-GPI) and the transmembrane isoform (CD160-TM) act as inducers of NK cells’ function, while the soluble form (sCD160) that is produced after cleavage of the GPI form interacts with HVEM and inhibits the function of CD4^+^T-cells [217], CD8^+^T-cells [218] and NK cells [219]. BTLA belongs to the Ig superfamily and contains an intracellular domain with membrane proximal ITIM and membrane-distal ITSM parts that are similar to those of PD-1 and CTLA-4. Upon BTLA/HVEM binding, inhibitory signals are conducted via phosphorylation of ITIMs [220]. In patients with metastatic melanoma, Haymaker et al. reported that BTLA/HVEM binding suppressed proliferation and cytokine production in BTLA^+^CD8^+^TILs [221]. In a study of 116 metastatic melanoma patients, Malissen et al. showed that high HVEM expression promotes the tumor’s evasion and impairs OS compared to individuals with low HVEM expression (37.3 months vs. 67.7 months, *p* = 0.0160) [222]. Furthermore, Gauci et al. observed significantly higher CD160-GPI positivity in specimens from melanoma lesions compared to benign human nevi and associated the release of sCD60 from melanoma cells with impaired NK function, immune escape and dissemination [219].

Regarding therapeutic implications, BTLA inhibition, in vitro, in combination with PD-1 and TIM-3 blockade reactivated dysfunctional melanoma antigen-specific CD8^+^T-cells, enhancing their proliferation and cytokine secretion [223]. In mouse models, co-inhibition of PD-1 and BTLA was found to be superior to anti-PD-1 monotherapy [224], supporting the testing of BTLA in trials of combinatorial ICI regimens. Icatolimab (JS004, TAB004) is a novel anti-BTLA mAb currently under investigation for patients with advanced solid malignancies, including melanoma, refractory to standard therapies in two phase-I trials, as monotherapy or in combination with PD-1 inhibition (NCT04137900 and NCT04773951, respectively). Among the 19 patients treated with icatolimab in the dose-expansion part of the NCT04137900 study, one melanoma patient that had previously progressed on nivolumab and BRAF/MEK inhibitors reached a PR with no grade 4/5 TRAEs [225]. Lastly, NCT04477876 is an observational study aiming to quantify CD160-TM/CD160-GPI expression in melanoma patients and to assess the therapeutic potential of anti-CD160-TM/CD160-GPI agonist mAbs.

## 10. B7-H3 (B7 Homolog 3 Protein, CD276)

B7-H3 comprises a type-I transmembrane protein that plays a complex immunoregulatory role, potentially exhibiting both co-stimulatory and co-inhibitory effects. Although B7-H3 mRNA is found in the majority of normal tissues, miRNA-mediated extensive post-transcriptional regulation largely suppresses B7-H3 protein expression [226]. Contrary to healthy tissues, an analysis of 9433 tumor samples from The Cancer Genome Atlas revealed significantly higher B7-H3 expression in 15 of 31 cancer types, including malignant melanoma [227]. The triggering receptors expressed in myeloid cells (TREM)-like transcript 2 (TLT2), IL-20 receptor subunit α (IL20Ra) and phospholipase A2 receptor 1 (PLA2R1) are proposed as the main B7-H3 receptors, although contradictory data exist [228]. TLT-2 is expressed in neutrophils, macrophages and B-cells as well as CD8^+^ and CD4^+^T-cells [229,230,231]. Regarding the functional significance of B7-H3/TLT2 interaction, conflicting insights have been published, as the studies of Hashiguchi et al. [229] and Chapoval et al. [232] reported enhancement of CD8^+^T-cell function and increased IL-2 and IFN-γ production, whereas later studies opposed engagement between B7-H3 and TLT2 [233,234,235]. In particular, Leitner et al. presented that B7-H3 significantly inhibits CD4^+^ and CD8^+^T-cell proliferation (*p* < 0.001, and *p* < 0.01, respectively) and markedly reduces cytokine production by these cells in the absence of B7-H3/TLT2 binding [233]. Similar findings were reported by Prasad et al. who described that B7-H3 downregulates NFAT, NF-kB and AP-1 transcriptional factors and consequently restricts T-cell function and effector cytokine secretion [236]. Further studies support the inhibitory effect of B7-H3 on T-cells [237,238]. For instance, B7-H3 was shown to hinder NK-cell functions and its targeting by anti-B7-H3 mAbs was found to effectively enhance NK-cell antitumor activity [239]. The IL20RA ligand of B7-H3 is not expressed on immune cells and, thus, B7-H3/IL20RA interaction indirectly restricts the immune response, shaping an immunosuppressive TME with increased PD-L1 expression, decreased numbers of CD4^+^ and CD8^+^T-cells and increased infiltration by MDSCs [240]. Binding between B7-H3 and PLA2R1 has also been reported by Cao et al.; however, its immunomodulatory effect is yet to be described [241].

Although the exact activity of B7-H3 remains inconclusive, robust evidence associates its expression with aggressive behavior and poor prognosis in several malignancies [242]. In melanoma, B7-H3 mRNA expression was shown to be significantly increased with the stage of melanoma and significantly associated with melanoma-specific survival in both stage-III and -IV melanoma patients; in addition, its expression was also related to migration and invasion [243]. In murine models, its inhibition was found to potentiate the killing potential of CD8^+^T-cells and NK cells and to significantly abrogate melanoma-cell proliferation [244]. Furthermore, B7-H3 expression was associated with decreased treatment response to chemotherapy and small-molecule inhibitors [245], while B7-H3 knockdown successfully restored sensitivity to such treatments [246]. Tekle et al. further demonstrated that the metastatic capacity of melanoma is enhanced by B7-H3 via altering metastasis-associated proteins, MMP-2 and STAT3 in murine models [247]. To block its metastatic propensity, several anti-B7-H3 modalities have been examined in vivo and in vitro against melanoma and other solid-tumor tissues, and, recently, anti-B7-H3 agents have also entered clinical testing [227,248,249] (Table 5). Interim results of a phase-I/II trial evaluating the combination of an anti-B7-H3 mAb, enoblituzumab (MGA271), and pembrolizumab in patients with advanced solid tumors yielded poor results in the melanoma cohort as only one of the 13 melanoma patients had a PR (ORR = 7.7%) (NCT02475213) [250]. Enoblituzumab was also tested along with ipilimumab in melanoma patients with disease progression under prior ICI in a completed phase-I study without published results (NCT02381314). Furthermore, CAR T-cell immunotherapy targeting B7-H3 for melanoma is currently being studied in three clinical trials (NCT05190185, NCT04483778, NCT04897321). Preliminary data from the ongoing STRIVE-02 trial (NCT04483778) in children and young adults with relapsed/refractory solid malignancies yielded encouraging anti-tumor activity and no dose-limiting toxicities [251]. Vobramitamab duocarmazine (MGC018) is an ADC of a humanized anti-B7-H3 mAb conjugated to a synthetic duocarmycin analogue [252], which is currently under investigation in two clinical trials in combination with lorigerlimab (anti-PD-1 and anti-CTLA-4 bispecific) (NCT05293496, NCT03729596). Results from NCT03729596 showed a manageable safety profile with early evidence of clinical activity in metastatic melanoma refractory to ≥two prior lines of ICI therapy [253].

## 11. Conclusions

More than a decade after the first approved ICI in melanoma, it is evident that the era of immunotherapy is still in its infancy. Beyond PD-1/PD-L1 and CTLA-4, several checkpoint molecules, as described above, have been identified and much more will come soon, while numerous immunotherapeutic agents (e.g., other ICIs, bsAbs) blocking their functional interactions are under investigation at different levels of research development. Following the gradual understanding of the complex crosstalk between melanoma cells and immune-system components, the focus has turned to unravelling immune pathways that work synergistically with the known checkpoint targets, with an aim to push the boundaries of immunotherapy resistance even further. In addition to the conventional biomarkers of TMB, neoantigen load and CD8^+^T-cell infiltration, a growing set of preclinical and clinical melanoma studies has provided valuable insights into novel TME-originated and blood-borne biomarkers that can predict response or resistance to immunotherapy. Comprehensive analyses of scRNA-Seq datasets have highlighted novel signatures that were positively correlated with ICI resistance in melanoma [254] while T-cell receptor clonality and TCR profiling, in melanoma patients, have shown greater diversification of pre-treatment circulating CD4^+^ or CD8^+^T-cell repertoires and were favorably correlated with longer survival times than patients with lower clone numbers [255,256]. In addition, baseline circulating DNA have recently become popular as a non-invasive indicator of tumor burden for real-time assessment of ICI response [257,258]. The detection of the high-sensitivity immune biomarkers behind melanoma resistance to ICIs and the individualized development of specific immunotherapeutic regimens to target specific melanoma-patient sub-cohorts according to their immunological profile are essential requirements for moving to the next level. More interventions on the immune response, including TILs, oncolytic viruses, and mRNA vaccines, are expected to make this future more convoluted; however, the moment of “personalized immunotherapy” is hopefully not too far away.

## Figures and Tables

**Figure 1 cancers-15-02718-f001:**
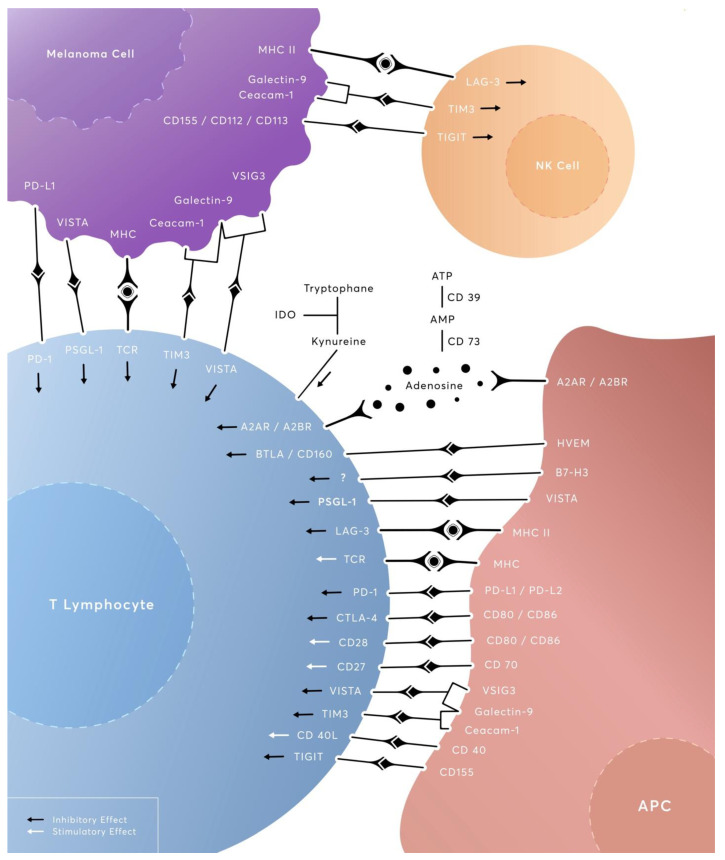
An overview of the complex checkpoint-mediated crosstalk between melanoma cells and immune effector cells.

**Table 1 cancers-15-02718-t001:** Ongoing clinical trials testing LAG-3-targeted modalities in melanoma.

Drug	Format	ClinicalTrials.gov Identifier	Phase	Indication	Intervention	Target
Relatlimab (BMS-986016)	Fully human IgG4κ monoclonal antibody	NCT05704933	I	Neoadjuvant therapy for surgically resectable melanoma brain metastases	Relatlimab + Nivolumab vs. Nivolumab + Ipilimumab	Anti-LAG-3 + Anti- PD-1 vs. Anti-PD-1 + Anti-CTLA-4
		NCT05629546	I	Advanced or metastatic melanoma, after progression on ICI therapy	Memory-like natural killer cells (autologous or allogeneic) with Relatlimab + Nivolumab	ML NKs with Anti-LAG-3 + Anti- PD-1
		NCT04935229	I	Metastatic uveal melanoma in the liver	Pressure-enabled hepatic artery infusion of SD-101, alone or in combination with Nivolumab, Nivolumab + Ipilimumab, or Relatlimab + Nivolumab	PEDD/HAI TLR9-agonist +/− Anti-PD-1, Anti-PD-1 + Anti-CTLA-4 or Anti-LAG-3 + Anti-PD-1
		NCT01968109	I/IIa	Advanced solid tumors, including melanoma that is immunotherapy-naïve and immunotherapy-experienced	Single-agent Relatlimab or Relatlimab + Nivolumab	Anti-LAG-3 or Anti-LAG-3 + Anti-PD-1
		NCT03978611	I/IIa	Unresectable or metastatic melanoma, after progression on anti-PD-1 ICI therapy	Relatlimab + Ipilimumab	Anti-LAG-3 + Anti-CTLA-4
		NCT02465060	II	Solid tumors, including melanoma, after progression ≥ one line of standard treatment or for which no agreed-upon treatment approach exists, including melanoma	Subprotocol Z1M (LAG-3 expression ≥ 1%); Relatlimab + Nivolumab	Anti-LAG-3 + Anti-PD-1
		NCT05704647	II	Active melanoma brain metastases	Relatlimab + Nivolumab	Anti-LAG-3 + Anti- PD-1
		NCT05428007	II	Unresectable clinical-stage-III–IV melanoma	Sarilumab, Ipilimumab and Relatlimab + Nivolumab	Anti-IL-6R, Anti-CTLA-4 and Anti-LAG-3 + Anti- PD-1
		NCT05418972	II	Neoadjuvant +/− adjuvant therapy for high risk, clinical-stage-II cutaneous melanoma	pre-surgery +/− post-surgery Relatlimab + Nivolumab	Anti-LAG-3 + Anti- PD-1
		NCT03743766	II	Metastatic melanoma naïve to prior immunotherapy in the metastatic setting	Nivolumab, Relatlimab, or Nivolumab + Relatlimab followed by Nivolumab + Relatlimab	Anti-PD-1, Anti-LAG-3 or Anti-LAG-3 + Anti-PD-1
		NCT04552223	II	Metastatic uveal melanoma	Relatlimab + Nivolumab	Anti-LAG-3 + Anti- PD-1
		NCT05077280	II	Metastatic uveal melanoma	Concurrent stereotactic radiotherapy with Relatlimab + Nivolumab	SBRT with Anti-LAG-3 + Anti-PD-1
		NCT03470922	II/III	Previously untreated metastatic or unresectable melanoma	Relatlimab + Nivolumab vs. Nivolumab	Anti-LAG-3 + Anti- PD-1 vs. Anti-PD-1
		NCT05625399	III	Previously untreated metastatic or unresectable melanoma	Relatlimab + Nivolumab	Anti-LAG-3 + Anti- PD-1
		NCT05002569	III	Adjuvant therapy for completely resected clinical-stage-III–IV melanoma	Relatlimab + Nivolumab vs. Nivolumab	Anti-LAG-3 + Anti- PD-1 vs. Anti-PD-1
Fianlimab (REGN3767)	Fully human IgG4κ monoclonal antibody	NCT03005782	I	Advanced malignancies, including melanoma	Fianlimab +/− Cemiplimab	Anti-LAG-3 +/− Anti-PD-1
		NCT05352672	III	Previously untreated unresectable locally advanced or metastatic melanoma	Fianlimab + Cemiplimab vs. Pembrolizumab or Cemiplimab	Anti-LAG-3 + Anti-PD-1 vs. Anti-PD-L1 or Anti-PD-1
		NCT05608291	III	Adjuvant therapy for completely resected high-risk melanoma	Fianlimab + Cemiplimab vs. Pembrolizumab	Anti-LAG-3 + Anti-PD-1 vs. Anti-PD-L1
LBL-007	Fully human IgG4κ monoclonal antibody	NCT04640545	I	Unresectable or metastatic melanoma	LBL-007 + Toripalimab or LBL-007 + Toripalimab and Axitinib Tablets	Anti-LAG-3 + Anti-PD-1 or Anti-LAG-3 + Anti-PD-1 + TKI
INCAGN02385	Humanized Fc-engineered IgG1κ monoclonal antibody	NCT04370704	I/II	Advanced melanoma	INCAGN02385 + INCAGN02390 or INCAGN02385 + INCAGN02390 + INCMGA00012	Anti-LAG-3 + Anti-TIM-3 or Anti-LAG-3 + Anti-TIM-3 + Anti-PD-1
RO7247669 (RG-6139)	Fc-inert IgG1-based bispecific antibody	NCT04140500	I	Advanced and/or metastatic solid tumors, including melanoma	single-agent RO7247669	Anti-(PD-1 × LAG-3)
		NCT05116202	Ib/II	Immunotherapy-naïve, stage-III/IV melanoma	Nivolumab + Ipilimumab or RO7247669 or Atezolizumab + Tiragolumab or RO7247669 + Tiragolumab	Anti-PD-1 + Anti -CTLA-4 or Anti-(PD-1 × LAG-3) or Anti-PD-L1 + Anti-TIGIT or Anti-(PD-1 × LAG-3) + Anti-TIGIT
		NCT05419388	II	Previously untreated unresectable or metastatic melanoma	Single-agent RO7247669	Anti-(PD-1 × LAG-3)
XmAb^®^22841 (Bavunalimab)	Fc-engineered half-IgG1κ/scFv bispecific antibody	NCT03849469	I	Advanced solid tumors including melanoma	XmAb22841 +/− Pembrolizumab	Anti-(CTLA-4 × LAG-3) +/− Anti-PD-L1
		NCT05695898	Ib/II	Metastatic melanoma refractory to prior ICI therapy with and without CNS disease	XmAb22841 + XmAb23104	Anti-(CTLA-4 × LAG-3) + Anti-(PD-1 × ICOS)
EMB-02	FIT-Ig^®^ bispecific antibody	NCT04618393	I/II	Advanced solid tumors, including locally advanced/metastatic melanoma with >1 prior therapy (PD-1/L1 +/− CTLA-4 ICI)	Single-agent EMB-02	Anti-(PD-1 × LAG-3)

**Table 2 cancers-15-02718-t002:** Ongoing clinical trials testing TIGIT-targeted modalities in melanoma.

Drug	Format	ClinicalTrials.gov Identifier	Phase	Indication	Intervention	Target
Tiragolumab (RG6058, MTIG7192A)	Fully human IgG1 monoclonal anitbody	NCT05483400	II	Anti-PD-1-resistant melanoma	Tiragolumab + Atezolizumab	Anti-TIGIT + anti-PD-1
		NCT05116202	II	Immunotherapy-naïve stage-III/IV melanoma	Tiragolumab + Atezolizumab or Tiragolumab + RO7247669	Anti-TIGIT + anti-PD-1 or anti-TIGIT + anti-PD1-LAG3
		NCT05060003	II	Circulating tumor DNA+ stage-II melanoma	Tiragolumab + Atezolizumab	Anti-TIGIT + anti-PD-1
		NCT03554083	II	High-risk stage-III melanoma	Tiragolumab + Atezolizumab	Anti-TIGIT + anti-PD-1
Domvanalimab (AB154)	Humanized IgG1 monoclonal antibody	NCT05130177	II	Anti-PD-1 relapsed/refractory melanoma	Domvanalimab + Zimberelimab	Anti-TIGIT + anti-PD-1
Vibostolimab (MK-7684)	Humanized IgG1 monoclonal antibody	NCT04303169	I/II	Pre-surgery stage-III melanoma	Vibostolimab + Pembrolizumab	Anti-TIGIT + anti-PD-1
		NCT04305054	I/II	Advanced melanoma	Vibostolimab + Pembrolizumab or Vibostolimab + Pembrolizumab+ Favezelimab	Anti-TIGIT + anti-PD-1 or anti-TIGIT + anti-PD-1 + anti-LAG3
		NCT04305041	I/II	Anti-PD-1 refractory melanoma	Vibostolimab + Pembrolizumab+ Quavonlimab	Anti-TIGIT + anti-PD-1 + anti-CTLA-4
		NCT05665595	III	High-risk stage-II–IV melanoma	Vibostolimab + Pembrolizumab vs. Pembrolizumab	Anti-TIGIT + anti-PD-1
EOS-448	Fully human IgG1 monoclonal antibody	NCT05060432	I/II	Advanced solid tumors	EOS-448 + Pembrolizumab or EOS-448 + Inupadenant or EOS-448 + Dostarlimab or EOS-448 + Dostarlimab+ Inupadenant	Anti-TIGIT + anti-PD-1 or anti-TIGIT+ anti-A2AR or anti-TIGIT + anti-PD-1 or anti-TIGIT + anti-PD-1 + anti-A2AR

**Table 3 cancers-15-02718-t003:** Ongoing clinical trials testing TIM-3-targeted modalities in melanoma.

Drug	Format	ClinicalTrials.gov	Phase	Indication	Intervention	Target
Cobolimab (TSR-022)	Humanized IgG4 monoclonal antibody	NCT02817633	I	Advanced solid tumors including melanoma	Cobolimab or Cobolimab + nivolumab or cobilimab + TSR-042 or cobolimab + TSR-042 and TSR-033 or cobolimab + TSR-042 and docetaxel or Cobolimab + TSR-042, pemetrexed and cisplatin or Cobolimab + TSR-042, pemetrexed and carboplatin	Anti-TIM-3 or anti-TIM-3 + anti-PD-1 or anti-TIM-3 + anti-PD-1 and anti-LAG-3 or anti-TIM-3 + anti-PD-1 and chemotherapy
		NCT04139902	II	Pre-surgery regionally advanced or oligometastatic melanoma	Cobolimab + Dostarlimab vs. Dostarlimab	Anti-TIM-3 + anti-PD-1 vs. anti-PD-1
Sabatolimab (MBG453)	Humanized anti-TIM-3 IgG4	NCT02608268	I/II	Advanced solid tumors including melanoma	Sabatolimab + PDR001 or Sabatolimab	Anti-TIM-3 + anti-PD-1 or anti-TIM-3
INCAGN02390	Fully human Fc-engineered IgG1κ monoclonal antibody	NCT04370704	I	Advanced solid tumors including melanoma	INCAGN02390 + INCAGN02385 or INCAGN02390 + INCAGN02385 + INCMGA00012	Anti-TIM-3 + anti-LAG-3 or anti-TIM-3 + anti-LAG-3 + anti-PD-1
RO7121661	CrossMabVH-VL bispecific antibody	NCT03708328	I	Advanced solid tumors including melanoma	RO7121661	Anti-PD-1/TIM-3
TQB2618	N/A	NCT05451407	Ib	Advanced melanoma	TQB2618 + Terprizumab	Anti-TIM-3 + anti-PD-1

**Table 4 cancers-15-02718-t004:** Ongoing clinical trials testing modalities targeting VISTA, IDO, CD27/CD70, A2AR and BTLA.

Target	Drug	Format	ClinicalTrials.gov Identifier	Phase	Indications	Intervention	Target
VISTA	KVA12123	Fully human IgG1 monoclonal antibody	NCT05708950	I/II	Advanced solid tumors	KVA12123 or KVA12123 + pembrolizuman	Anti-VISTA or anti-VISTA + anti-PD-1
	W0180	Humanized IgG1 monoclonal antibdoy	NCT04564417	I	Locally advanced or metastatic solid tumors	W0180 or W0180 + pembrolizumab	Anti-VISTA or anti-VISTA + anti-PD-1
	CI-8993	Fully human IgG2 monoclonal antibody	NCT04475523	I	Locally advanced or metastatic solid tumors	CI-8993	Anti-VISTA
	HMBD-002	Humanized IgG4	NCT05082610	I	Advanced solid tumors	HMBD-002 or HMBD-002 + pembrolizumab	Anti-VISTA or anti-VISTA + anti-PD-1
IDO	IO102-IO103	IDO/PD-L1 dual peptide vaccine	NCT05280314	II	Pre-surgery and post-surgery stage IIIB, IIIC, or IIID melanoma or oligometastatic IV melanoma	IO102-IO103 + pembrolizumab	IDO/PD-L1 vaccine + anti-PD-1
			NCT05155254	III	Previously untreated unresectable or metastatic melanoma	IO102-IO103 + pembrolizumab or pembrolizumab	IDO/PD-L1 vaccine + anti-PD-1 or anti-PD-1
CD27	Varlimumab (CDX-1127)	Fully human IgG1 monoclonal antibody	NCT03617328	I/II	Stage-II–IV melanoma	Varlimumab + 6MHP + Montanide ISA-51 + polyCLC or 6MHP + Montanide ISA-51 + polyCLC	Anti-CD27 + multipeptide vaccine + local adjuvant + local adjuvant or multipeptide vaccine + local adjuvant + local
CD70	Anti-hCD70 CAR T-cells	Anti-hCD70 CAR T-cells	NCT02830724	I/II	Unresectable solid tumors	Anti-hCD70 CAR + cyclophosphamide + flutarabine + aldesleukin	Anti-hCD70 CAR + alkylating agent + antimetabolite + recombinant IL-2
Adenosine A2A receptor	Inupadenant	A2A receptor antagonist	NCT05117177	I	Advanced solid tumors	Inupadenant	A2Ar antagonist
			NCT05060432	I/II	Advanced solid tumors	Inupadenant + EOS-448 or Inupadenant + dostarlimab or Inupadenant + dostarlimab + EOS-448	A2Ar antagonist + anti-TIGIT or A2Ar antagonist + anti-PD-1 or A2Ar antagonist + anti-TIGIT + anti-PD-1
BTLA	Icatolimab (JS004, TAB004)	Humanized IgG4 monoclonal antibody	NCT04773951	I	Advanced solid tumors	Icatolimab	Anti-BTLA
			NCT04137900	I	Advanced solid tumors	Icatolimab + toripalimab or icatolimab	Anti-BTLA + anti-PD-1 or anti-BTLA

**Table 5 cancers-15-02718-t005:** Ongoing clinical trials testing B7-H3-targeted modalities in melanoma.

Drug	Format	ClinicalTrials.gov Identifier	Phase	Indication	Intervention	Target
Vobramitamab duocarmazine (MGC018)	Humanized IgG1 Anti-B7-H3 antibody-drug conjugate	NCT01391143	I	Advanced solid tumors, including melanoma	Enoblituzumab	B7-H3 ADCC
		NCT03729596	I/II	Advanced solid tumors, including melanoma	Vobramitamab duocarmazine or Vobramitamab duocarmazine + retifanlimab	Anti-B7-H3 ADC or Anti-B7-H3 ADC + Anti-PD-1
TAA06	B7-H3-Targeted CAR-T Cells	NCT05190185	I	Advanced solid tumors, including melanoma	TAA06 injection	B7-H3-Targeted CAR-T Cells
4-1BBζ B7H3-EGFRt-DHFR	B7-H3-Targeted CAR-T Cells	NCT04483778	I	Solid tumors, including melanoma, in children and young adults	4-1BBζ B7H3-EGFRt-DHFR or 4-1BBζ B7H3-EGFRt-DHFR + 4-1BBζ CD19-Her2tG	B7-H3-Targeted CAR-T Cells
B7-H3-CAR T cells	B7-H3-Targeted CAR-T Cells	NCT04897321	I	Recurrent/refractory B7-H3 positive solid tumors, including melanoma, after lymphodepleting chemotherapy	B7-H3-CAR T cells + Fludarabine + Cyclophosphamide + MESNA	B7-H3-Targeted CAR-T Cells + chemotherapy + chemotherapy protective agent

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
