# Peer review of "Beyond CTLA-4 and PD-1 Inhibition: Novel Immune Checkpoint Molecules for Melanoma Treatment"

_cancers, 2023, doi:10.3390/cancers15102718_

Round 1
Reviewer 1 Report
The authors have reviewed the literature on the current and novel immune checkpoint inhibitors (ICI) for melanoma treatment. They present the currently approved ICI (anti-PD-1 and anti-CTLA-4) and their limitations. The main focus of the study is a comprehensive overview of all the other options targeting novel checkpoint molecules such as LAG-3, TIGIT, TIM-3, etc… to overcome resistance and disease relapse.
For each target, the authors present the function and the mechanism of action of the inhibitors followed by a comprehensive analysis of the different pre-clinical and clinical studies with a list of the different ongoing trials.
Overall, this is a well-presented study.
Minor comments
Introduction, Sentence between lines 64-66 – Needs a reference
All Tables: could be sorted by the Phase of the trials as it doesn’t seem to be any logical order to the current presentation. A “target” column instead of the the “class” column” will help simplifying the text.
Table 3 – The targets are missing for 1 of the trials with Cobolimab.
Line 348 – LPS needs to be defined. Lipopolysaccharides?
Table 4 – Trial NCT05280314. Which stages?
Author Response
We would like to thank the reviewer for the general positive comments and the constructive feedback on our manuscript, which has helped us to improve the quality of our article.
Comment 1: Introduction, Sentence between lines 64-66 – Needs a reference. Authors’ reply: A reference has been added to support the statement made.
Comment 2: All Tables: could be sorted by the Phase of the trials as it doesn’t seem to be any logical order to the current presentation. A “target” column instead of the “class” column will help simplify the text. Authors’ reply: We re-organized the tables accordingly, as requested.
Comment 3: Table 3 – The targets are missing for 1 of the trials with Cobolimab. Author’s reply: The targets omitted here have been added.
Comment 4: Line 348 – LPS needs to be defined. Lipopolysaccharides? Authors’ reply: The full term “Lipopolysaccharides” has been added before LPS (now within brackets) for clarification.
Comment 5: Table 4 – Trial NCT05280314. Which stages? Authors’ reply: The melanoma stages investigated in the trial in question have been added to the respective table cell.
Reviewer 2 Report
Ziogas et al. review the important topical of novel checkpoint inhibitors in the treatment of melanoma. This manuscript is topical, particularly in light of the recent approval of anti-LAG-3 therapy for advanced melanoma. Overall, the manuscript is well-written and comprehensive. However, in its current form, I feel it requires minor edits to address a few concerns before it would be acceptable to publication.
1) The authors state in the introduction that ICI therapy reverses T cell exhaustion (Line 59). Current understanding of the mechanism of checkpoint inhibitors, particularly anti-PD-1 therapy, is that in enhances the proliferation of populations of precursor exhausted cells (Tpex) but that it does not reverse the actual process of T cell exhaustion. These data have been expounded in many papers, including but not limited to: PMID: 35624209, PMID: 35961204, PMID: 35978192, PMID: 36971604. The authors should modify the text to clarify this important distinction as it represents an important paradigm shift in the understanding of checkpoint inhibitor therapy.
2) Similarly, the role of Neo-antigens in driving T cell exhaustion remains controversial. The authors should acknowledge this and that other tumor antigens can contribute to the chronic TCR stimulation which drives T cell dysfunction and exhaustion
3)Most of the references are very current but a recent publication suggesting the intracellular mechanism of LAG-3 function has been published. It should be cited and added to the discussion of LAG-3 function. Paper: PMID: 35437325
Author Response
We would like to thank the reviewer for the general positive comments and the constructive feedback on our manuscript, which has helped us to improve the quality of our article.
Comment 1: The authors state in the introduction that ICI therapy reverses T cell exhaustion (Line 59). Current understanding of the mechanism of checkpoint inhibitors, particularly anti-PD-1 therapy, is that it enhances the proliferation of populations of precursor exhausted cells (Tpex) but that it does not reverse the actual process of T cell exhaustion. These data have been expounded in many papers, including but not limited to: PMID: 35624209, PMID: 35961204, PMID: 35978192, PMID: 36971604. The authors should modify the text to clarify this important distinction as it represents an important paradigm shift in the understanding of checkpoint inhibitor therapy. Authors’ reply: We modified the introduction text accordingly to include the indicated data.
Comment 2: Similarly, the role of Neo-antigens in driving T cell exhaustion remains controversial. The authors should acknowledge this and that other tumor antigens can contribute to the chronic TCR stimulation which drives T cell dysfunction and exhaustion. Authors’ reply: We proceeded to text adjustment to reflect the abovementioned statement.
Comment 3: Most of the references are very current but a recent publication suggesting the intracellular mechanism of LAG-3 function has been published. It should be cited and added to the discussion of LAG-3 function. Paper: PMID: 35437325.Authors’ reply: The recent data on the proposed intracellular mechanism of LAG-3 function has been included in the respective section of the manuscript.
Reviewer 3 Report
This review is nicely written to summarize current checkpoint molecules roles' in melanoma treatment.
Overall, topic is very well introduced and organized. Adequate references used, even though more could be added but it is hard to be most comprehensive as the area is highly active.
My only critique would be the lack of biomarkers discussion in the manuscript. This review would benefit from, at least with a paragraph, in the conclusion part discussing the current biomarker studies. Clinical studies are now mostly focusing on biomarker studies to improve the outcome of trials. Monotherapies versus combo approaches are considered based on biomarkers that target the tumor immune microenvironment signatures.
Quality of written English is very good -except for a few points. It could be edited by a professional editor but it is not essential.
p1, line 10 - ":" after the first word "Although" needs to be taken off
p1, line 18 - replace the word "study" with "review"
p 2, line 46 - reference needed after 2nd sentence ending ... by 2027.
p 2, line 51 - open name "Overall Survival" for OS should be inserted
p 2, line 60 - "Up to now" could be replaced by "As of now"
p 2, line 60 - insert (FDA) after Food and Drug Administration
Author Response
We would like to thank the reviewer for the general positive comments and the constructive feedback on our manuscript, which has helped us to improve the quality of our article.
Comment 1: My only critique would be the lack of biomarkers discussion in the manuscript. This review would benefit from, at least with a paragraph, in the conclusion part discussing the current biomarker studies. Clinical studies are now mostly focusing on biomarker studies to improve the outcome of trials. Monotherapies versus combo approaches are considered based on biomarkers that target the tumor immune microenvironment signatures.
Authors’ reply: We created a new paragraph in the section of Conclusions to address the biomarker talk and reflect its significance in immunotherapy strategies and personalized treatment. Nevertheless, as the biomarker field is ever-growing and profoundly active in literature, we had to go over only a few representative examples, as further details are beyond the scope of the current paper.
Comment 2: p1, line 10 - ":" after the first word "Although" needs to be taken off. Authors’ reply: We stand corrected.
Comment 3: p1, line 18 - replace the word "study" with "review". Authors’ reply: We inserted the word “review” replacing “study”.
Comment 4: p 2, line 46 - reference needed after 2nd sentence ending ... by 2027. Authors’ reply: A reference was added to support the statement made.
Comment 5: p 2, line 51 - open name "Overall Survival" for OS should be inserted. Authors’ reply: Open name “Overall Survival” was inserted instead.
Comment 6: p 2, line 60 - "Up to now" could be replaced by "As of now". Authors’ reply: We made the proposed correction.
Comment 7: p 2, line 60 - insert (FDA) after Food and Drug Administration. Authors’ reply: Abbreviation FDA within brackets was inserted following “Food and Drug Administration”.